# T-3DGS: Removing Transient Objects for 3D Scene Reconstruction

## Abstract

Transient objects in video sequences can significantly degrade the quality of 3D scene reconstructions. To address this challenge, we propose T-3DGS, a novel framework that robustly filters out transient distractors during 3D reconstruction using Gaussian Splatting. Our framework consists of two steps. First, we employ an unsupervised classification network that distinguishes transient objects from static scene elements by leveraging their distinct training dynamics within the reconstruction process. Second, we refine these initial detections by integrating an off-the-shelf segmentation method with a bidirectional tracking module, which together enhance boundary accuracy and temporal coherence. Evaluations on both sparsely and densely captured video datasets demonstrate that T-3DGS significantly outperforms state-of-the-art approaches, enabling high-fidelity 3D reconstructions in challenging, real-world scenarios.

## 1 Introduction

Novel view synthesis and 3D scene reconstruction from multiple 2D images or videos are critical, rapidly evolving areas in computer vision. Neural Radiance Fields (NeRF) (Mildenhall et al., 2021) and 3D Gaussian Splatting (3DGS) (Kerbl et al., 2023) have shown remarkable improvements in novel view synthesis on complex scenes. NeRF implicitly represents the scene as a volumetric function, and 3DGS explicitly represents it as a set of 3D Gaussians. Both approaches produce high-quality, realistic images. There are multiple follow-up works for diverse downstream applications, including 3D scene reconstruction (Wang et al., 2021; Li et al., 2023; Guédon & Lepetit, 2024), 3D synthesis (Poole et al., 2022; Wynn & Turmukhambetov, 2023; Tang et al., 2025), semantic and language integration into 3D representations (Siddiqui et al., 2023; Kerr et al., 2023; Shi et al., 2024).

Both 3D Gaussian Splatting and NeRF optimize 3D scene reconstruction using photometric losses. High-quality results are achieved under the assumption that the captured scene is completely static and does not include any *distractors*, such as moving objects (i.e. transient objects), shadows, lightning changes, etc. In real-world scenarios, this assumption can hardly be satisfied. Ignoring distractors during scene optimization results in undesired blurring effects and floating artifacts. At the same time, removing such distractors from the captured recordings is challenging and limits the widespread usage of NeRF and 3DGS. Additionally, we would like to identify semi-transient objects in recordings and remove them from the scene. We define a semi-transient object as an object that has both dynamic and static states during the capturing process, e.g. a pushed chair stops after some time and becomes a fully static object.

We introduce T-3DGS, a novel approach for 3D static scene reconstruction from monocular video in uncontrolled settings. Our method includes an unsupervised transient detector and a transient mask propagation framework. Relying solely on image residuals for transient identification is unreliable due to issues such as appearance changes and color similarity to the background (Ren et al., 2024; Sabour et al., 2024). To address this issue, we develop a divergence-based technique on top of the uncertainty modeling approach (Kulhanek et al., 2024) to detect transients. It helps improve mask accuracy and significantly reduces misclassifications of transient objects.

Our experiments show that concurrent works (Sabour et al., 2024; Ungermann et al., 2024) fail to remove semi-transient distractors (Fig. 1). We introduce a mask propagation framework for extracting object-aware masks that improve consistency in the case of semi-transient distractors. Our

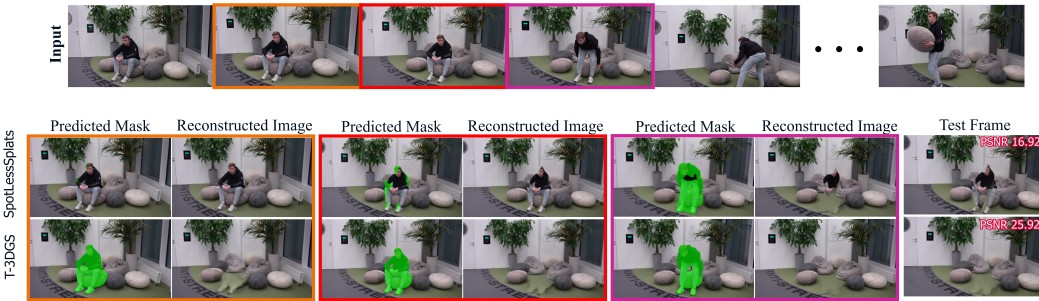

Figure 1: Existing state-of-the-art methods, such as SpotLessSplats Sabour et al. (2024), often struggle to correctly identify transient and semi-transient objects, leading to artifacts in 3D scene reconstruction. Our proposed *T-3DGS* method accurately detects all transient distractors, generates clean masks, and propagates them across frames. By effectively masking transient objects, *T-3DGS* enables high-fidelity novel view synthesis and significantly improves reconstruction quality from real-world image and video sequences.

method remains robust to all types of distractors. Additionally, we present the novel *T-3DGS dataset* with challenging scenes featuring semi-transient and slow-moving objects. Evaluations on both casual scenes (Ren et al., 2024; Sabour et al., 2023) and our dataset show our method outperforms state-of-the-art approaches in reconstruction quality.

Our key contributions, which together ensure consistent detection and removal of transient objects for improved 3D reconstruction, include:

- Generalized uncertainty modeling for efficient transient object identification;
- A divergence-based approach that leverages semantic consistency between reference and reconstructed frames for identifying transient objects;
- A robust video object segmentation module that tracks objects across varying frame rates and semi-transient behaviors;
- A challenging new dataset featuring diverse scenes with semi-transient distractors and slow-moving objects;
- State-of-the-art performance on benchmark datasets for robust static scene reconstruction.

## 2 RELATED WORK

We provide a brief review of the works on Neural Radiance Fields and 3D Gaussian Splatting with a focus on removing non-static distractors in the scene.

Neural Radiance Fields (NeRFs) (Mildenhall et al., 2021) are widely adopted methods for high-quality scene reconstruction and novel view synthesis of 3D scenes. The seminal work 3D Gaussian Splatting (Kerbl et al., 2023) employs Gaussian primitives to model scenes instead of relying on continuous volumetric representations. This method has recently gained popularity as a faster alternative to NeRFs.

**Handling Distractors in NeRFs.** NeRF-W (Martin-Brualla et al., 2021) and RobustNeRF (Sabour et al., 2023) are two pioneering works approaching the problem in a similar way. NeRF-W reconstructs both static background and transients combined with a data-dependent uncertainty field. RobustNeRF utilizes Iteratively Reweighted Least Squares for transient object identification and removal. Both methods rely on color residual supervision and frequently misclassify transient objects and backgrounds that share similar colors. Additionally, they both require careful hyper-parameters tuning. NeRF On-the-go (Ren et al., 2024) utilizes DINOv2 features (Oquab et al., 2023) to identify and eliminate distractors by predicting uncertainties through a shallow MLP and can deal with more complex scenes than RobustNeRF.

**Extracting Features from Vision Foundation Models**. Vision Foundation Models (VFMs) are trained on large-scale data, enabling strong generalization to unseen domains or novel tasks. Task-agnostic models trained through self-distillation, like DINO (Caron et al., 2021; Oquab et al., 2023), learn features that can be generalized for multiple vision tasks.

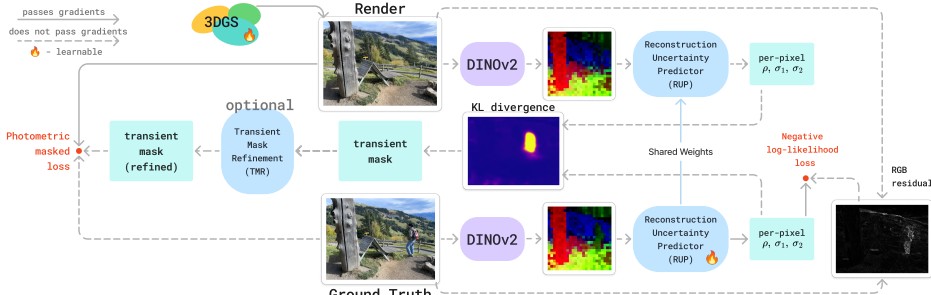

Figure 2: *Overview of the Proposed T-3DGS Architecture.* We introduce a modified version of 3D Gaussian Splatting, incorporating a masked loss term $\mathcal{L}_{\text{masked}}$ as described in Eq. 13. In each iteration, we start by rendering a reconstruction of a randomly sampled reference image. We compute residuals, along with DINOv2 features from ground truth and reference images. These features are then fed to our *RUP* model to predict the per-pixel covariance matrix for both images. We calculate binary masks based on the divergence of these distributions (as specified in Eq. 10). Subsequently, we compute the likelihood as described in Eq. 7 and update the parameters of the *RUP* model via backpropagation, as indicated by the dashed lines. Additionally, for some scenes, we incorporate a SAM-based mask refiner module (*TMR*), which further enhances the consistency and sharpness of the masks.

**Video Object Segmentation**. The goal of semi-supervised VOS is to identify when an object appears for the first time and then track it throughout the video. Several recent approaches based on transformers (Cheng et al., 2023; 2024) have been proposed. However, current methods suffer from mask inconsistencies, particularly when objects disappear and reappear in the video. Additionally, these methods assume that the input has a high frame rate, and they become unstable when the frame rate is low. In our work, we address these shortcomings.

**Handling Distractors in 3DGS.** Several works address the training of 3DGS on unconstrained, in-the-wild photo collections. Robust 3DGS (Ungermann et al., 2024) proposes a self-supervised approach to identify transient distractors by utilizing image residuals and leveraging a pre-trained segmentation network to produce object-aware masks. SpotLessSplats (Sabour et al., 2024) proposes a method to identify transient objects by utilizing pre-computed feature maps from a foundation model (Tang et al., 2023) coupled with a robust optimization of 3DGS. These and other works (Dahmani et al., 2024; Zhang et al., 2024; Xu et al., 2024; Ungermann et al., 2024; Sabour et al., 2024) suffer from: 1) the need for hyper-parameter tuning, such as threshold parameters; 2) inaccurate prediction of transient masks across the video; and 3) reliance on image residuals, leading to the false detection of transients, as shown in Fig. 1. In our approach, we aim to address the limitations of the current works by identifying transients more accurately and consistently across video frames.

## 3 METHOD

We propose a novel approach to reconstructing static scenes from unconstrained videos that contain dynamic objects, utilizing 3D Gaussian Splatting (3DGS). Our method, illustrated in Fig. 2, introduces two key components designed to handle dynamic objects effectively: (1) **reconstruction uncertainty predictor (*RUP*)**, and (2) **transient mask refiner (*TMR*)**. The transient area detection component, implemented through our transient mask learning predictor, identifies regions containing dynamic objects by predicting per-pixel probabilities using semantic features. The transient mask refiner improves transient detections in both spatial and temporal domains by leveraging SAM2 Ravi et al. (2024) to propagate transient masks across multiple frames.

**3D Gaussian Splatting.** Our method is based on 3DGS (Kerbl et al., 2023). Given a set of posed images $\{I_n\}_{n=1}^{N}, I_n \in \mathbb{R}^{H \times W \times C}$, 3DGS represents a 3D scene as a set of anisotropic Gaussians $\{\mathcal{G}_i\}$, where each Gaussian is represented by its position (mean) $\mu_i$, a positive semi-definite covariance matrix $\Sigma_i$, an opacity $\alpha_i$, and a view-dependent appearance component (color) parametrized by spherical harmonics (SH) (Ramamoorthi & Hanrahan, 2001). Each 3D Gaussian is projected onto the image plane through a differentiable rasterization process to render an image from a specific viewpoint. The 3DGS representation is learned through optimization of Gaussian parameters via stochastic gradient descent.

## 3.1 UNCERTAINTY MODELING

**Uncertainty Prediction.** Given the input images $\{I_n\}_{n=1}^{N}$, the goal is to optimize the unsupervised reconstruction uncertainty predictor *RUP* through 3DGS reconstruction to identify transient distractors without explicit supervision as shown in Fig. 2. Following prior research (Martin-Brualla et al., 2021; Ren et al., 2024; Kulhanek et al., 2024) we employ uncertainty modeling techniques, although with significant modifications. *RUP* is trained to identify transient objects without explicit supervision, purely from the reconstruction objectives. As in WildGaussians (Kulhanek et al., 2024) every iteration we update both Gaussian Splatting and *RUP* weights and 1) we detach masks when updating Gaussian Splatting, 2) we detach reconstructed images when updating *RUP*. This is done to avoid trivial solutions where every pixel is masked as dynamic.

Following (Kulhanek et al., 2024; Sabour et al., 2024), we reformulate the transient detection problem as a semantic feature classification task. This approach leverages pre-trained foundation models to extract rich semantic features from images. By doing so, it enables our system to make decisions based on high-level semantic understanding, rather than relying solely on color information.

**Feature Extraction.** For each training iteration, we extract DINOv2 features (Oquab et al., 2023) from both the input image $I$ and the corresponding rendering $\hat{I}$, producing feature maps $f$ and $\hat{f}$, respectively. We choose DINOv2 for several key reasons: (1) its self-supervised training enables robust semantic understanding without class-specific biases, (2) it demonstrates strong performance in distinguishing object boundaries and semantic regions even for previously unseen objects. These features serve as a robust foundation, enabling *RUP* to make accurate decisions about scene dynamics without explicit supervision.

**Transient 2D Uncertainty Modeling** As previously discussed, most methods detect transient objects by utilizing reconstruction errors. For example, NeRF On-the-go (Ren et al., 2024) considers per pixel RGB residuals, defined as errors between pairs of pixels:

$$R = ||\hat{I} - I||_2. \tag{1}$$

It assumes that the residuals follow a normal distribution:

$$p(R|\sigma) = \frac{1}{\sqrt{2\pi\sigma^2}} \exp\left(-\frac{R^2}{2\sigma^2}\right). \tag{2}$$

Therefore, we can obtain the negative log likelihood:

$$\mathcal{L}_u = \frac{R^2}{2\sigma^2} + \log\sigma + \frac{\log 2\pi}{2}. \tag{3}$$

Although the approach is reasonable, RGB residuals lack robustness. In particular, high-frequency objects often result in high reconstruction errors, producing misclassifications. Similarly, dynamic objects with colors similar to the background may be classified as static. While DSSIM or DINOv2 cosine distance can mitigate some errors, they introduce their own limitations. In particular, DINOv2 cosine distance, while highly robust, suffers from low resolution. Upsampling models, such as FeatUP (Fu et al., 2024), can address this issue, though they introduce upsampling artifacts.

This motivates a new multivariate formulation of uncertainty modeling. Let the per-pixel residual be a 2-dimensional vector:

$$R = \begin{bmatrix} R_1 \\ R_2 \end{bmatrix}, \tag{4}$$

where $R_1$ and $R_2$ correspond to different similarity metrics. We chose DINOv2 cosine distance defined like in WildGaussians, except we use features upscaled with FeatUP (Fu et al., 2024) as $R1$ and DSSIM as $R2$. We consider a multivariate normal distribution with zero mean and covariance matrix $\Sigma$:

$$p(R) = \frac{1}{(2\pi)\sqrt{|\Sigma|}} \exp\left(-\frac{1}{2}R^T\Sigma^{-1}R\right), \tag{5}$$

where the covariance matrix is given by

$$\Sigma = \begin{bmatrix} \sigma_1^2 & \rho\sigma_1\sigma_2 \\ \rho\sigma_1\sigma_2 & \sigma_2^2 \end{bmatrix}. \tag{6}$$

The negative log-likelihood function becomes:

$$\mathcal{L}_u = -\log p(R) = \frac{1}{2}R^T\Sigma^{-1}R + \frac{1}{2}\log|\Sigma| + \log 2\pi. \quad (7)$$

In contrast to previous works (Martin-Brualla et al., 2021; Ren et al., 2024; Kulhanek et al., 2024) we predict three parameters — $\sigma_1$, $\sigma_2$, $\rho$ instead of a single $\sigma$. This allows us to combine information from several residuals.

Similarly as in previous methods, we assume that residuals follow a normal distribution. Although a better derivation that utilizes more complicated and accurate distributions might be preferable, we went with a more heuristic but simple approach.

We train a neural network that takes DINOv2 features from a reference image as input and makes a per pixel prediction of $\Sigma$, and use our likelihood term in Eq. 7 as a loss function. We predict $\sigma_i$ using a softplus nonlinearity, and $\rho$ using a tanh nonlinearity to avoid undesirable values. In practice, this approach can be numerically unstable (e.g., due to $\Sigma$ being ill-conditioned when $\sigma_i \rightarrow 0$ or $\rho \rightarrow \pm 1$). In our experience, the introduction of clamping and normalization layers into the architecture of the neural network mostly solved this problem.

Given that our training objective is considerably more challenging than the one-dimensional modeling of WildGaussians (Kulhanek et al., 2024), our model requires a larger architecture. However, this also offers an advantage over previous methods, as we can add upscale layers to make our prediction denser without sacrificing local/nonlocal smoothing. The details of the architecture are provided in the Supplementary Material.

**Binary Mask.** One approach to obtaining a binary mask using the modeled uncertainty is to set a threshold on one of the predicted values or define a new criterion:

$$M = \mathbb{I}(f(\sigma_1, \sigma_2, \rho) > C), \quad (8)$$

where $\mathbb{I}$ is the indicator function, $f(\sigma_1, \sigma_2, \rho)$ is a chosen criterion and $C$ is a threshold chosen as a hyperparameter.

However, this methodology has notable limitations when applied to the reconstruction of geometrically complex static structures. In such cases, even static objects may produce substantial residuals, leading to their misclassification as dynamic elements and their subsequent masking.

We note that, even though we train our *RUP* only on reference images, we can also obtain a per-pixel uncertainty prediction $\hat{\Sigma}$ using an image reconstructed by Gaussian Splatting. Because our model relies on the semantic information of DINOv2 features, we should expect it to make a similar prediction in the static areas and a different one in areas corresponding to the dynamic objects. To estimate this discrepancy, we calculate the Kullback-Leibler (KL) divergence $D_{KL}(\mathcal{N}(0,\Sigma)||\mathcal{N}(0,\hat{\Sigma}))$, which, following a standard proof (Duchi, 2007), takes following form:

$$D_{KL}(\mathcal{N}(0,\Sigma)||\mathcal{N}(0,\hat{\Sigma})) = \frac{1}{2}(tr(\hat{\Sigma}^{-1}\Sigma) - ln(\frac{|\Sigma|}{|\hat{\Sigma}|}) - 2). \quad (9)$$

Fig. 3 illustrates how this approach reduces false classifications in static regions. Unlike previous methods that obtain masks by estimating uncertainty, we instead utilize divergence. This allows us to incorporate additional information to enhance the consistency of our masks. Hence, our binary masks are obtained based on the new criterion:

$$M = \mathbb{I}(D_{KL} > C). \quad (10)$$

In particular, the use of KL divergence for thresholding purposes was explored in several papers (Yadav & Singh, 2016; Jabari et al., 2019). Although we diverge significantly from this method, there is a strong similarity – thresholding divergence between two images (in particular, input image and background-only image) is simply more robust than thresholding photometric errors.

On the other hand, while we directly follow several approaches, in particular Kulhanek et al. (2024), utilizing divergence instead of photometric errors adds an additional layer of robustness by incorporating information from both ground truth and reconstructed images. Coincidentally, this is also

a convenient way of turning complicated distributions (even multivariate ones) into a single number that can be simply thresholded.

**Training Stability.** During 3DGS optimization, there are periods when renders may be unreliable, particularly at the beginning of training and after each opacity reset. Building on (Kulhanek et al., 2024), we address this by implementing two key strategies. First, we delay the start of *RUP* training until the 3DGS optimization has completed its first 500 iterations, ensuring the initial scene reconstruction has reached sufficient quality. Second, after each opacity reset, we temporarily pause the *RUP* optimization for 250 iterations while we keep training 3DGS, allowing the reconstruction to stabilize before resuming transient detection. We also use the scheduled sampling technique from SpotLessSplats (Sabour et al., 2024).

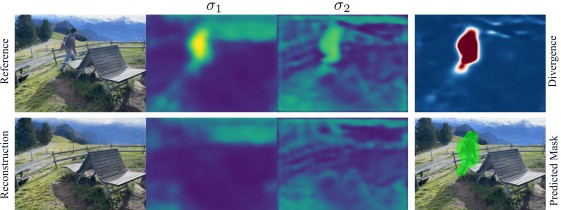

Figure 3: During the initial stages of reconstruction, *RUP* predicts high uncertainty in challenging regions such as backgrounds or high-frequency details. However, since *RUP* relies exclusively on semantic information, calculating the divergence between reference uncertainty $\Sigma$ and reconstructed uncertainty $\hat{\Sigma}$ effectively suppresses these artifacts. Areas with divergence values above the threshold are highlighted in red, while the final predicted transient mask by *RUP* is shown in green.

**Mask Dilation.** We also dilate our masks, depending on the resolution of the scene. This dilation step serves multiple purposes, primarily covering shadows and reflections caused by objects. These modifications ensure robust training and accurate detection of transient objects in diverse dynamic scenes.

### 3.2 TRANSIENT MASK REFINEMENT

Our transient area detection pipeline is robust for current benchmarks when transient objects change their positions from frame to frame. However, for semi-transient objects, which may not change positions for some frames, it fails and masks only parts of the video when they are dynamic. To address this issue, we introduce a mask propagation process that refines transient masks into temporally consistent masks across the entire video sequence through refinement and propagation.

**Spatial Refinement.** We use the Segment Anything Model (SAM) (Kirillov et al., 2023) to refine our transient maps, $P_i$, into more precise masks, $M'_i$. For each connected component $C_i^k$ in $P_i$, we sample up to ten points as prompts for SAM, leveraging its ability to generate high-quality segmentations from sparse inputs to extract a set of object-aware masks $M'_i = \{M'^j_i\}_{j=1}^{L_i}$, where $L_i$ is the number of predicted masks for image $I_i$. Due to potential inaccuracies in the boundaries of our masks, some sampled points might occasionally fall on the background rather than the object itself (e.g. a point sampled between the legs of a person). To address this, we filter the predicted masks based on their local coverage score:

$$CS_{\text{local},i} = \frac{|P_i \cap M'^j_i|}{|M'^j_i|}. \tag{11}$$

We keep masks that satisfy $CS_{\text{local},i} > \lambda_{\text{cov}}^{\text{ref}}$, forming the refined set $M'_i = \{M'^j_i | CS_{\text{local},i} > \lambda_{\text{cov}}^{\text{ref}}\}$.

**Temporal Refinement.** To address potential false negatives, we propagate the refined masks, $\{M'_i\}_{i=1}^N$, throughout the video using SAM2 (Ravi et al., 2024) to obtain more consistent masks, $\{M_i\}_{i=1}^N$. Our propagation process consists of three stages: 1) Forward Propagation: Iterating from the first frame to the last, propagating the segmentation masks forward. 2) Backward Propagation: Iterating from the last frame to the first, propagating information from future frames backward. 3) Final Propagation: A final first-to-last pass, considering both past and future frames as context, which helps to resolve temporal inconsistencies.

To manage computational resources efficiently, we introduce a memory size parameter, $N_m$, which limits the number of frames considered during propagation. At each step, we maintain and use segmentations from $N_m$ nearest frames, balancing temporal consistency with memory constraints.

During propagation, we manage mask intersections to ensure consistent segmentation. For any pair of masks $M_i^l$ and $M_i^m$ where $IoU(M_i^l, M_i^m) > \lambda_{merge}$, we merge them into a single mask, assigning the lower of the two original labels to maintain consistency.

**Dynamic Object Filtration.** To filter out false positive transients and ensure robust detection, we introduce the Stability Ratio (SR) metric, which combines spatial overlap accuracy and temporal consistency. For each detected object, the SR is calculated as $SR = \frac{1}{N} \sum_{i=1}^{N} (R_i \cdot CS_{\text{global},i})$, where $N$ is the number of valid frames, $R_i$ is the mean value of the absolute difference between ground truth and rendered images within the masked region in frame $i$, and $CS_{\text{global},i} = |P_i \cap M_i| / |M_{\text{max}}|$ is the global coverage score. Here, $P_i$ represents the prompt mask in frame $i$, $M_i$ is the segmentation mask, and $M_{\text{max}}$ is the maximum size of the object mask across all frames. This global score evaluates the object's consistency relative to its largest observed size. A frame is considered valid and contributes to the SR calculation only if its local coverage score (Eq. 11) exceeds the validation threshold $\lambda_{\text{cov}}^{\text{val}}$. Objects with $SR$ below a threshold $\lambda_{SR}$ are filtered out as potential false detections. This dual coverage score system ensures that objects maintain both spatial accuracy through local coverage and temporal consistency through global coverage and difference image values.

**Artifact-Free Reconstruction.** 3DGS tends to generate artifacts ("floaters") near the camera, particularly in challenging regions like those identified by transient masks. We address this issue through depth-aware regularization.

We render the depth $D$ for each pixel using alpha compositing, similar to color rendering: $D = \sum_{i=1}^{M} T_i \alpha_i d_i$, where $d_i$ is the depth value of the i-th Gaussian, $T_i$ is the accumulated transmittance, and $\alpha_i$ is the opacity value. To suppress floating artifacts while preserving sharp depth discontinuities at object boundaries, we apply anisotropic total variation (TV) regularization to the rendered depth map: $\mathcal{L}_{\text{depth}} = \text{mean}(|\nabla_x D|) + \text{mean}(|\nabla_y D|)$, where $\nabla_x$ and $\nabla_y$ are spatial gradients in $x$ and $y$ directions respectively.

### 3.3 Masked Gaussian Splatting Optimization

The final step involves training the Gaussian Splatting model with the obtained masks $\{M_i\}_{i=1}^{N}$ for transients. Let $M_i$ be the binary mask for frame $i$, defined as:

$$M_i(x,y) = \begin{cases} 1 & \text{if } (x,y) \text{ is in an occluded area,} \\ 0 & \text{if } (x,y) \text{ is in a static area,} \end{cases} \tag{12}$$

where $(x,y)$ represents pixel coordinates in the image. We apply binary dilation to $M_i$ for $N_e$ iterations, yielding $M_i^*$. This operation creates a buffer zone around detected dynamic objects, improving the robustness of our static scene reconstruction. The final loss for 3DGS is:

$$\mathcal{L}_{\text{masked}} = \lambda_{\text{SSIM}} \cdot L_{\text{SSIM}}(I_i \odot \overline{M}_i^*, \hat{I}_i \odot \overline{M}_i^*) + \lambda_{\text{L1}} \cdot \left\| \overline{M}_i^* \odot (I_i - \hat{I}_i) \right\|_1 + \lambda_{\text{depth}} \mathcal{L}_{\text{depth}}, \tag{13}$$

where $I_i$, $\hat{I}_i$ are reference images and their reconstructions, $\odot$ is the Hadamard product, $\|\cdot\|_1$ is L1 norm, $L_{SSIM}$ is a structural similarity loss, $\overline{M}_i^*$ is a negation of $M_i^*$ that represents a static background and $\lambda_{\text{SSIM}}$, $\lambda_{\text{L1}}$ and $\lambda_{\text{depth}}$ are weighting factors.

This formulation allows the model to focus on static scene elements, effectively handling dynamic objects in the reconstruction process. By integrating these steps, our method reconstructs static scenes robustly from unconstrained videos while effectively handling transient distractors.

## 4 Experiments

We evaluate our proposed T-3DGS model on various datasets captured in uncontrolled settings and filled with diverse distractors. We perform qualitative and quantitative comparisons against state-of-the-art methods. Finally, we provide an ablation study of architectural and loss function choices. We discuss the limitations of the proposed method in the Supplementary Material.

**Datasets.** We evaluate our model on three challenging datasets. The *NeRF On-the-go dataset* (Ren et al., 2024) contains four outdoor and two indoor sparsely captured scenes with different levels of occlusion (from 5% to over 30%) and minimal appearance changes. The *RobustNeRF*

| | Mountain | | Fountain | | Corner | | Patio | | Spot | | Patio High | | Mean | |
|---|---|---|---|---|---|---|---|---|---|---|---|---|---|---|
| | PSNR ↑ | SSIM ↑ | PSNR ↑ | SSIM ↑ | PSNR ↑ | SSIM ↑ | PSNR ↑ | SSIM ↑ | PSNR ↑ | SSIM ↑ | PSNR ↑ | SSIM ↑ | PSNR ↑ | SSIM ↑ |
| NeRF On-the-go (Ren et al., 2024) | 20.15 | 0.64 | 20.11 | 0.61 | 24.22 | 0.81 | 20.78 | 0.75 | 23.33 | 0.79 | 21.41 | 0.72 | 21.67 | 0.72 |
| 3DGS (Kerbl et al., 2023) | 19.40 | 0.66 | 19.96 | 0.66 | 20.90 | 0.71 | 17.48 | 0.70 | 20.77 | 0.69 | 17.29 | 0.60 | 19.30 | 0.67 |
| Robust3DGS (Ungermann et al., 2024) | 16.97 | 0.61 | 18.18 | 0.59 | 23.47 | 0.85 | 21.33 | 0.85 | 22.61 | 0.88 | 21.81 | 0.82 | 20.73 | 0.77 |
| WildGaussians (Kulhanek et al., 2024) | 20.77 | 0.70 | 20.74 | 0.67 | 25.79 | 0.88 | 21.77 | 0.85 | 24.39 | 0.88 | 22.36 | 0.80 | 22.64 | 0.80 |
| SpotLessSplats (Sabour et al., 2024) | 21.25 | 0.66 | 20.49 | 0.63 | 25.59 | 0.85 | 21.13 | 0.80 | 24.13 | 0.78 | 22.18 | 0.76 | 22.46 | 0.75 |
| **Ours** | 21.11 | **0.71** | **20.94** | **0.69** | **26.46** | **0.90** | **21.95** | **0.87** | **25.78** | **0.90** | **22.76** | **0.83** | **23.17** | **0.82** |

Table 1: Quantitative comparison on the *On-the-go* dataset (Ren et al., 2024).

| | Lab 1 | | Lab 2 | | Library | | Anti-Stress | | Office | | Mean | |
|---|---|---|---|---|---|---|---|---|---|---|---|---|
| | PSNR ↑ | SSIM ↑ | PSNR ↑ | SSIM ↑ | PSNR ↑ | SSIM ↑ | PSNR ↑ | SSIM ↑ | PSNR ↑ | SSIM ↑ | PSNR ↑ | SSIM ↑ |
| 3DGS (Kerbl et al., 2023) | 24.49 | 0.91 | 20.42 | 0.87 | 20.08 | 0.89 | 20.45 | 0.86 | 26.96 | 0.94 | 22.48 | 0.89 |
| Robust3DGS (Ungermann et al., 2024) | 25.35 | 0.93 | 24.74 | 0.92 | 24.33 | 0.93 | 22.95 | 0.91 | 28.52 | **0.96** | 25.18 | 0.93 |
| WildGaussians (Kulhanek et al., 2024) | 25.71 | 0.92 | 23.68 | 0.91 | 24.65 | 0.92 | 21.69 | 0.89 | 28.89 | 0.95 | 24.92 | 0.92 |
| SpotLessSplats (Sabour et al., 2024) | 25.28 | 0.91 | 24.63 | 0.90 | 24.11 | 0.91 | 22.22 | 0.90 | 28.08 | 0.92 | 24.86 | 0.91 |
| 4D-GS (Wu et al., 2024) | 20.16 | 0.77 | 15.26 | 0.64 | 21.14 | 0.87 | 21.69 | 0.85 | 20.11 | 0.72 | 19.67 | 0.77 |
| Easi3R (Chen et al., 2025) | 24.08 | 0.90 | 22.85 | 0.90 | 26.64 | 0.95 | 26.37 | 0.94 | 25.94 | 0.92 | 25.18 | 0.92 |
| **Ours w/o TMR** | 25.77 | 0.93 | 24.67 | 0.92 | 24.67 | 0.93 | 24.07 | 0.92 | 29.36 | 0.95 | 25.71 | 0.93 |
| **Ours w/ TMR** | **27.76** | **0.96** | **25.54** | **0.93** | **28.25** | **0.97** | **29.01** | **0.96** | **29.85** | **0.96** | **28.08** | **0.96** |

Table 2: Quantitative comparison on the *T-3DGS* dataset.

*dataset* (Sabour et al., 2023) contains five indoor scenes with unintentional changes during the capture process. These changes include transient objects that appear and disappear without a consistent temporal order, as well as dynamic objects (e.g., floating balloons). Additionally, we introduce our novel *T-3DGS dataset*. The dataset contains 5 densely captured indoor scenes. Generally, dynamic objects in our videos are walking people and various small objects. However, unlike previous datasets, all scenes incorporate challenging cases, including transient, semi-transient, and slow-moving objects.

**Baselines.** We compare our model against vanilla 3D Gaussian Splatting (Kerbl et al., 2023) and the current state-of-the-art method, SpotLessSplats (Sabour et al., 2024). We further include WildGaussians (Kulhanek et al., 2024) and Robust3DGS (Ungermann et al., 2024) as baselines. In order to compare our method to Easi3R (Chen et al., 2025), we demonstrate results for 3DGS scenes trained with Easi3R masks. We also compare our method to 4D-GS (Wu et al., 2024) by reconstructing scenes and filtering out dynamic gaussians by a movement threshold. To compare different models, we use commonly used PSNR, SSIM (Wang et al., 2004), and LPIPS metrics for evaluation. LPIPS metrics are reported in the Supplementary Material.

**Implementation details.** All our experiments are conducted in accordance with the training setup from the official 3DGS implementation. We train our models for 30K iterations, using the Adam optimizer with a learning rate of 1e-3 for the *RUP*. The depth regularization loss $\mathcal{L}_{\text{depth}}$ is activated after the first 500 iterations.For the experiments with mask propagation, we first train the *RUP* for 7000 iterations. At that point, we pause the training and propagate the obtained transient masks. Subsequently, we initiate a new training procedure using the propagated masks, keeping all other parameters the same as the original training setup. We dilate all our masks by 10 pixels, except for the Patio scene, where we use the original mask due to its low resolution.

## 4.1 QUANTITATIVE COMPARISONS

We evaluate our model on all three datasets. We report results on *On-the-go* and *T-3DGS* datasets in Tab. 6 and 7, respectively, and we move the evaluation results of *RobustNeRF* dataset to the Supplementary Material as it presents the least challenge. As shown in Tab. 6 and 7, our method generally outperforms current SOTA methods. In particular, our method is robust to changes in distant and high-frequency details. In Tab. 6 we run our method directly on masks predicted by *RUP* module without mask propagator.

While current SOTA methods struggle to detect semi-transient objects (Tab. 7), our proposed transient network *RUP* achieves higher performance by minimizing false predictions. The integration of the SAM-based mask propagation *TMR* module further enhances our results in scenes containing semi-transient objects, providing more accurate and reliable reconstructions.

## 4.2 QUALITATIVE COMPARISONS

For qualitative evaluation, we compare our method to SpotLessSplats (Sabour et al., 2024), Robust3DGS (Ungermann et al., 2024), and WildGaussians (Kulhanek et al., 2024). Fig. 4 and 5 demonstrate that our method minimizes false negatives and effectively detects transients. For example, in the On-the-go dataset, most methods struggle with high-frequency details and distant objects, as these elements are typically reconstructed more slowly than the rest of the scene, leading to inac-

curacies in RGB residual-based approaches. However, due to our robust loss function, such artifacts are largely eliminated from our dynamic maps. Notably, SpotLessSplats uses features obtained from higher-resolution images, while we extract features at a lower resolution, the same resolution used for training 3DGS.

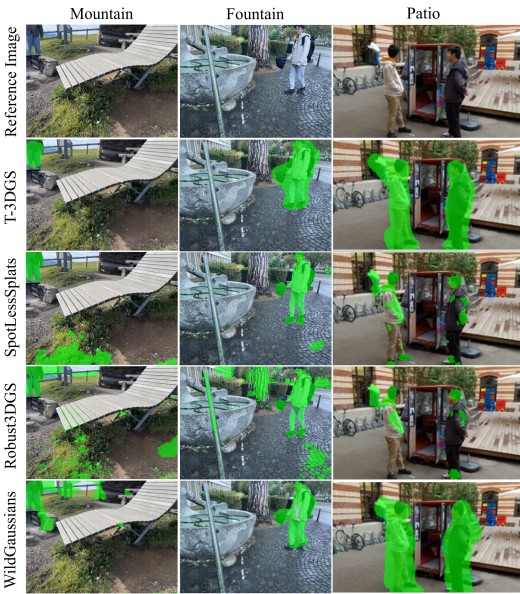

Figure 4: Qualitative results on the *On-the-go* dataset. Our method outperforms existing approaches in detecting transient objects. Predicted transient masks are shown in green.

For our *T-3DGS* dataset, we additionally utilize the SAM-based mask propagation module to propagate object-aware masks for semi-transient objects, as shown in Fig. 5. Although most methods would theoretically benefit from this technique, our masks are of higher quality and result in fewer incorrect detections. Applying mask propagation to other methods may introduce error propagation, as demonstrated in the Supplementary Material.

### 4.3 ABLATION STUDY

We present ablation results in Table 3 for the *On-the-go* dataset, excluding the Patio scene due to its low resolution. We evaluate our method under the following conditions: (1) without mask dilation, (2) without mask dilation and $\mathcal{L}_{\text{depth}}$, and (3) with both components enabled. Additionally, we report results obtained with ground truth masks while separately disabling $\mathcal{L}_{\text{depth}}$ and mask dilation by 10 pixels. Even when using ground truth masks, dilation noticeably enhances performance. This contradicts the assumptions made by NeRF-HuGS (Chen et al., 2024) and Robust3DGS (Ungermann et al., 2024), as exact masks do not yield optimal performance metrics. Furthermore, mask dilation aids *RUP* training by ensuring that all transient objects are fully covered.

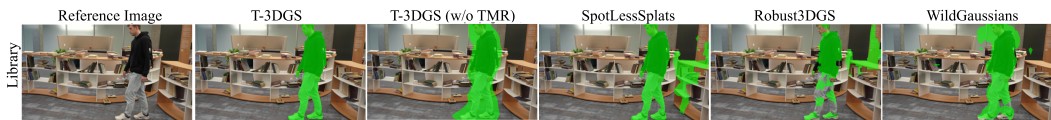

Figure 5: Qualitative results on the *T-3DGS* dataset. Our method produces cleaner transient masks and further refines them using the (*TMR*) module.

## 5 CONCLUSION

In this work, we have presented the novel *T-3DGS* method for 3D scene reconstruction using Gaussian Splatting by effectively filtering out foreground dynamic distractors from input videos. By integrating an unsupervised classification network with bivariate uncertainty modeling, KL divergence regularization, and a mask propagation strategy, our method achieves superior temporal coherence and boundary accuracy. Evaluations on both sparsely and densely captured datasets confirm significant improvements over state-of-the-art approaches.

|  | *On-the-go dataset* | |
|---|---|---|
|  | PSNR ↑ | SSIM ↑ |
| GT masks w/o $\mathcal{L}_{\text{depth}}$ | 22.84 | 0.82 |
| GT masks w/ $\mathcal{L}_{\text{depth}}$ and dilation | 23.43 | 0.81 |
| Ours w/o dilation and $\mathcal{L}_{\text{depth}}$ | 22.60 | 0.80 |
| Ours w/o dilation | 22.88 | 0.80 |
| **Ours (full)** | 23.41 | 0.81 |

Table 3: We evaluate the importance of each component of our method on the *On-the-go* dataset. We report the average performance across all scenes.

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

## A LIMITATIONS

We use features upscaled by FeatUP (Fu et al., 2024) to compute cosine distance, and while it is better than simple bilinear interpolation, it is relatively slow and gives fairly noisy results. Utilizing alternative ways to measure per pixel errors could improve both speed and accuracy of the method. Additionally, the temporal refinement process is constrained by a memory window of $N_m$ frames, which means that if an object disappears for more than $N_m$ frames and then reappears, it will be treated as a new instance with a different label. This can lead to inconsistent tracking and potentially affect the filtering process, especially for semi-transient objects that may temporarily leave the scene. Furthermore, our current filtering approach using global coverage scores may incorrectly filter out valid dynamic objects that undergo significant size changes, such as objects moving towards or away from the camera, or those experiencing perspective changes. We leave it for future work.

## B ADDITIONAL IMPLEMENTATION DETAILS

In Sec. 3.2, for mask filtering and refinement, we set $\lambda_{cov}^{ref} = 0.7$ for initial mask refinement and $\lambda_{cov}^{val} = 0.7$ for validation during object filtration. For temporal refinement, we set the memory size parameter $N_m = 10$, which controls the number of frames considered during mask propagation. For the final mask dilation step, we perform $N_e = 5$ iterations of binary dilation. In addition, the mask merging threshold $\lambda_{merge}$ is set to 0.9, and the stability ratio threshold $\lambda_{SR}$ to 0.08.

Our model consists of repeating blocks. We first use bilinear interpolation to increase the resolution of our features by two. We then apply a simple 3 by 3 convolutional layer that also decreases feature size by a factor of two. We then apply layer normalization followed by the GELU non-linearity. We repeat this sequence three times. After that we project our features with 1 by 1 convolution to obtain logits. We use softplus for $\sigma_1$, $\sigma_2$ and tanh for $\rho$. The normalization layer is crucial for improving the numerical stability that arises due to matrix $\Sigma$ being potentially ill-conditioned.

Our experiments were conducted with an RTX 6000 GPU. It takes around 40 minutes to train a scene using our method.

## C SOCIETAL IMPACTS

T-3DGS has the potential to substantially improve the reliability of 3D reconstruction technologies used in domains such as urban planning and the preservation of cultural heritage. However, its ability to isolate and remove humans from reconstructions can raise concerns about privacy and the erasure of social context in documentation and surveillance applications. Moreover, biases specific to vision foundation models can manifest themselves in our method, too.

| | Android | | Statue | | Crab (1) | | Crab (2) | | Yoda | | Mean | |
|---|---|---|---|---|---|---|---|---|---|---|---|---|
| | PSNR ↑ | SSIM ↑ | PSNR ↑ | SSIM ↑ | PSNR ↑ | SSIM ↑ | PSNR ↑ | SSIM ↑ | PSNR ↑ | SSIM ↑ | PSNR ↑ | SSIM ↑ |
| NeRF On-the-go (Ren et al., 2024) | 23.50 | 0.75 | 21.58 | 0.77 | - | - | - | - | 29.96 | 0.83 | - | - |
| 3DGS (Kerbl et al., 2023) | 23.51 | 0.81 | 21.35 | 0.84 | 30.39 | 0.94 | 31.53 | 0.92 | 29.80 | 0.92 | 27.32 | 0.89 |
| Robust 3DGS (Ungermann et al., 2024) | 24.40 | 0.83 | 22.10 | 0.85 | 34.41 | 0.96 | 32.99 | 0.93 | 32.62 | 0.93 | 29.30 | 0.90 |
| WildGaussians (Kulhanek et al., 2024) | 24.89 | 0.83 | 22.69 | 0.87 | 30.16 | 0.93 | 31.11 | 0.91 | 30.50 | 0.91 | 27.87 | 0.89 |
| SpotLessSplats (Sabour et al., 2024) | 24.45 | 0.79 | 22.50 | 0.80 | 35.45 | 0.95 | 33.29 | 0.94 | 33.55 | 0.94 | 29.85 | 0.88 |
| **Ours** | 25.10 | 0.84 | 22.90 | 0.87 | 34.25 | 0.95 | 33.85 | 0.93 | 32.45 | 0.93 | 29.71 | 0.90 |

Table 4: Quantitative comparison on the *RobustNeRF* dataset (Sabour et al., 2023).

| | Anti-Stress | | Lab (1) | | Lab (2) | | Library | | Office | | Mean | |
|---|---|---|---|---|---|---|---|---|---|---|---|---|
| | PSNR ↑ | SSIM ↑ | PSNR ↑ | SSIM ↑ | PSNR ↑ | SSIM ↑ | PSNR ↑ | SSIM ↑ | PSNR ↑ | SSIM ↑ | PSNR ↑ | SSIM ↑ |
| WildGaussians w/o TMR | 21.69 | 0.89 | 25.71 | 0.92 | 23.68 | 0.91 | 24.65 | 0.92 | 28.89 | 0.95 | 24.92 | 0.92 |
| WildGaussians w/ TMR | 24.07 | 0.92 | 24.65 | 0.92 | 24.84 | 0.93 | 28.32 | 0.97 | 29.75 | 0.96 | 26.33 | 0.94 |
| **Ours w/ TMR** | 28.79 | 0.97 | 27.71 | 0.95 | 25.42 | 0.93 | 28.34 | 0.97 | 29.87 | 0.96 | 28.03 | 0.96 |

Table 5: Evaluation of WildGaussians with *TMR* module on the *Transient-3DGS* dataset.

# D   T-3DGS DATASET

We provide a temporary anonymous link to our dataset. We will publish the dataset and the code for working with the data under a license that at a minimum allows free non-commercial use.

# E   ADDITIONAL EVALUATIONS

We evaluate our method on the *RobustNeRF* dataset (Sabour et al., 2023). As shown in Tab. 4 our method generally outperforms 3DGS (Kerbl et al., 2023), Robust 3DGS (Ungermann et al., 2024), WildGaussians (Kulhanek et al., 2024), and shows similar performance compared to Spot-LessSplats (Sabour et al., 2024). We run our method directly on masks predicted by *RUP* module without mask propagator (*TMR*). Overall, the dataset does not appear to be sufficiently challenging to differentiate between the methods.

We also provide LPIPS metrics for datasets mentioned in the paper. Tab.6 and Tab. 7 demonstrate that our method provides competitive results, although we find that this metric can be slightly inconsistent and hard to interpret.

# F   ADDITIONAL COMPARISONS WITH EASI3R

Recenty, several methods that rely on DUSt3R-like models have seen some popularity, most notably Easi3R (Chen et al., 2025). One of their main advantage is that they do not require additional inputs (e.g optical flow). However, we have discovered that this type of models generally does not generalize well in cases where optical flow is not applicable. We provide additional comparisons on the On-The-Go dataset. As shown in Tab. 8, Easi3R (Chen et al., 2025) performs similarly (if not worse) with vanilla 3DGS (Kerbl et al., 2023). This is in sharp contrast to previous comparisons in Tab. 7, where Easi3R (Chen et al., 2025) is extremely close to the state-of-the-art results.

While Easi3R (Chen et al., 2025) offers better inference speed, ultimately it performs poorly in cases where optical flow is not applicable.

# G   ADDITIONAL EXPERIMENTS WITH TMR MODULE

Even though our proposed *TMR* module leverages SAM2 to propagate the transient masks, we would like to emphasize that our method enables mask propagation spatially and temporally consistent, thereby providing more accurate and reliable reconstruction. Table 5 presents an evaluation of

| | Mountain | Fountain | Corner | Patio | Spot | Patio High | Mean |
|---|---|---|---|---|---|---|---|
| NeRF On-the-go (Ren et al., 2024) | 0.26 | 0.31 | 0.19 | 0.22 | 0.19 | 0.24 | 0.24 |
| 3DGS (Kerbl et al., 2023) | **0.21** | **0.19** | 0.24 | 0.20 | 0.32 | 0.36 | 0.25 |
| Robust3DGS (Ungermann et al., 2024) | 0.31 | 0.32 | 0.10 | **0.07** | 0.12 | **0.17** | 0.19 |
| WildGaussians (Kulhanek et al., 2024) | 0.23 | 0.21 | 0.09 | **0.07** | 0.10 | **0.17** | 0.15 |
| SpotLessSplats (Sabour et al., 2024) | 0.24 | 0.24 | 0.12 | 0.08 | 0.18 | 0.20 | 0.18 |
| **Ours** | 0.22 | 0.21 | 0.12 | 0.10 | 0.12 | **0.17** | 0.16 |

Table 6: LPIPS metrics on the *On-the-go* dataset (Ren et al., 2024).

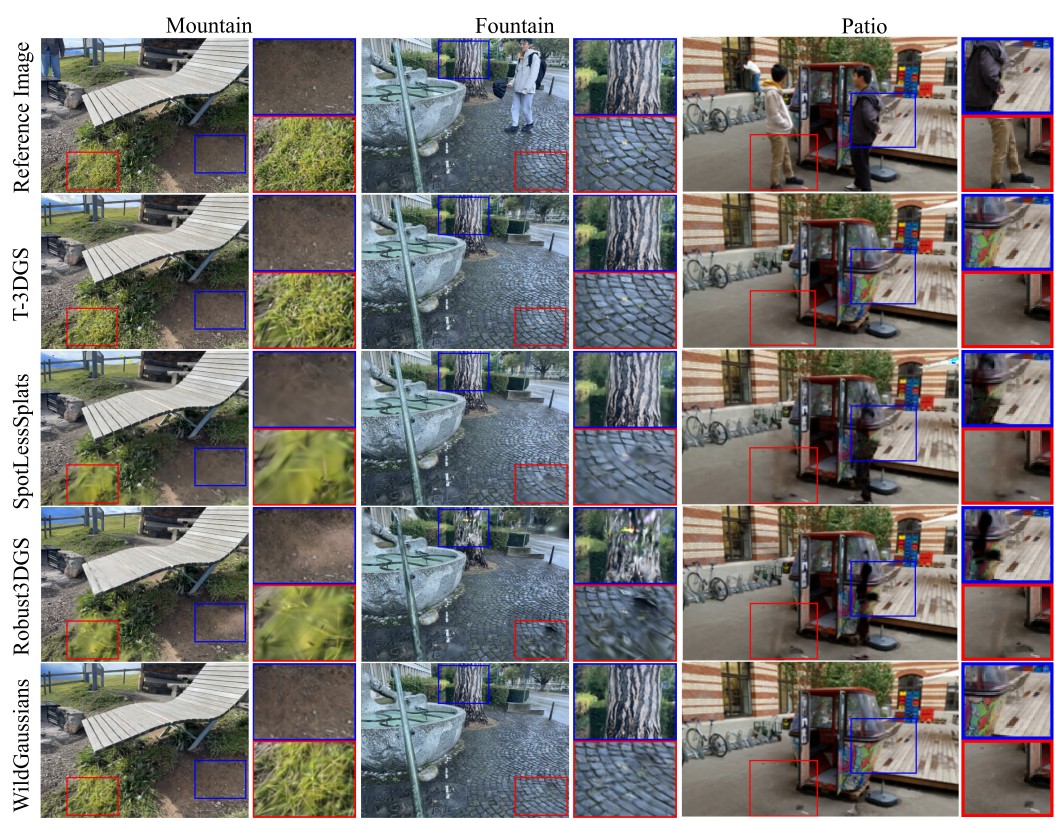

Figure 6: Qualitative results on the *On-the-go* dataset using the training frames. Our method produces higher-quality renderings without artifacts.

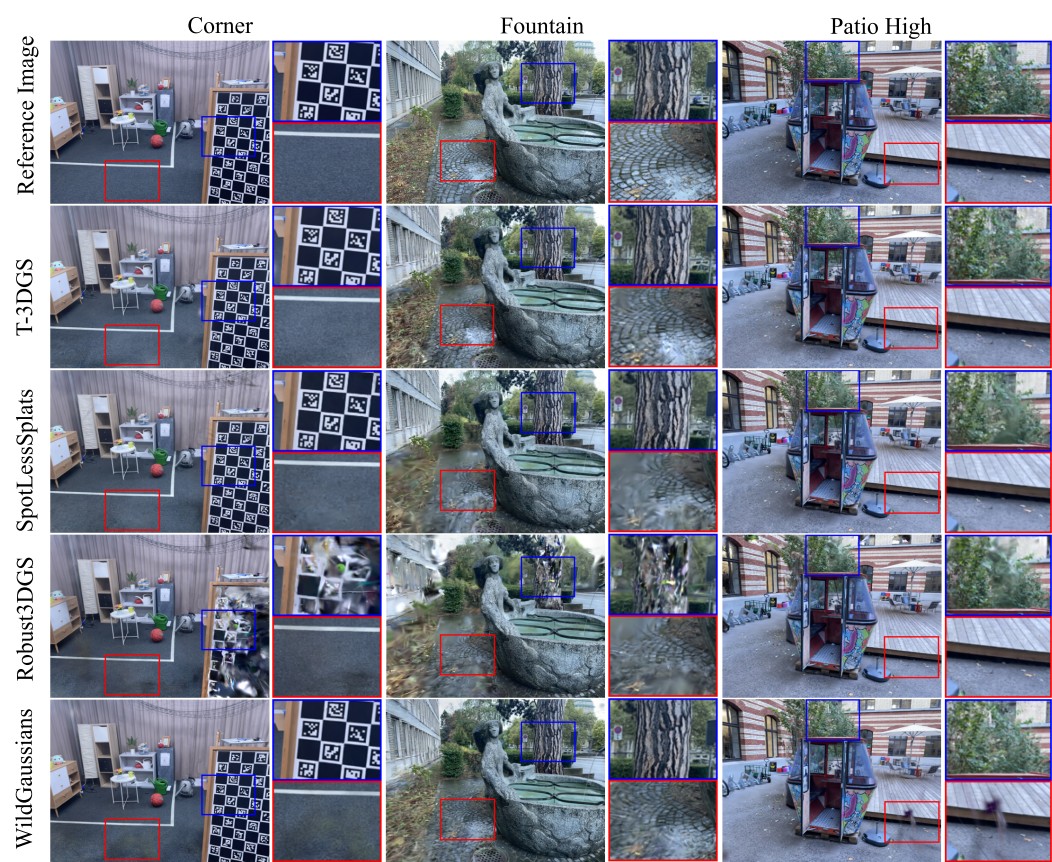

Figure 7: Qualitative results on the *On-the-go* dataset using the testing frames. Our method produces higher-quality renderings without artifacts.

| | Lab 1 | Lab 2 | Library | Anti-Stress | Office | Mean |
|---|---|---|---|---|---|---|
| Robust3DGS (Ungermann et al., 2024) | 0.09 | 0.10 | 0.08 | 0.10 | 0.06 | 0.09 |
| WildGaussians (Kulhanek et al., 2024) | 0.08 | 0.11 | 0.09 | 0.09 | 0.04 | 0.08 |
| SpotLessSplats (Sabour et al., 2024) | 0.08 | 0.09 | 0.09 | 0.10 | 0.05 | 0.08 |
| 4DGS (Wu et al., 2024) | 0.21 | 0.32 | 0.14 | 0.16 | 0.24 | 0.21 |
| Easi3R (Chen et al., 2025) | 0.11 | 0.12 | 0.06 | 0.07 | 0.09 | 0.09 |
| **Ours w/o TMR** | 0.09 | 0.10 | 0.08 | 0.09 | 0.05 | 0.08 |
| **Ours w/ TMR** | **0.02** | **0.06** | **0.02** | **0.02** | **0.02** | **0.03** |

Table 7: LPIPS metrics on the *T-3DGS* dataset.

| | Mountain | | Fountain | | Corner | | Spot | | Patio High | | Mean | |
|---|---|---|---|---|---|---|---|---|---|---|---|---|
| | PSNR ↑ | SSIM ↑ | PSNR ↑ | SSIM ↑ | PSNR ↑ | SSIM ↑ | PSNR ↑ | SSIM ↑ | PSNR ↑ | SSIM ↑ | PSNR ↑ | SSIM ↑ |
| 3DGS (Kerbl et al., 2023) | 19.40 | 0.66 | 19.96 | 0.66 | 20.90 | 0.71 | 20.77 | 0.69 | 17.29 | 0.60 | 19.30 | 0.67 |
| Easi3R (Chen et al., 2025) | 19.33 | 0.66 | 18.55 | 0.62 | 24.94 | 0.89 | 21.60 | 0.83 | 21.30 | 0.81 | 21.14 | 0.76 |
| **Ours** | **21.11** | **0.71** | **20.94** | **0.69** | **26.46** | **0.90** | **25.78** | **0.90** | **22.76** | **0.83** | **23.17** | **0.82** |

Table 8: Additional quantitative comparison on the *On-the-go* dataset (Ren et al., 2024) against Easi3R (Chen et al., 2025).

the reconstruction quality of WildGaussians with our *TMR* module. First, we obtained the transient masks using WildGaussians. Then, we propagate them through our *TMR* module. Finally, we reconstruct the scenes based on the transient masks obtained in the previous step. Our evaluation shows that our method produces higher-quality results for most scenes, with comparable performance in the remaining ones. Our method, with the *TMR* module, outperforms WildGaussians with the *TMR* module on Anti-Stress, Lab (1), Lab (2) scenes. The TMR module generally enhances the reconstruction quality of the original WildGaussians, but it is limited because of the false positive transient detections that come from WildGaussians itself. Furthermore, we note that the hyperparameters of our *TMR* module are highly dependent on the dataset rather than the model. That makes our *TMR* module robust across the different methods.

## H QUANTITATIVE MASK COMPARISONS

As models like SAM2 (Ravi et al., 2024) have gained popularity, many papers, rather than predicting good masks, have shifted focus towards predicting a good prompt for a segmentation model. Those prompts are primarily used to identify objects of interest and therefore reducing False Positives and increasing True Negatives becomes significantly more important, given that small errors can propagate though the entire scene, significantly degrading quality of classification (as discussed in the previous section). Unfortunately most commonly used metrics don't provide a good picture of those misclassifications. In order to quantitative estimate them, we have compared our method against WildGaussians (Kulhanek et al., 2024) and Easi3R (Chen et al., 2025),Tab. 9 provides mean Accuracy, Precision, Recall, Specificity, and Fallout for dynamic masks on the T-3DGS scenes, where Specificity is defined as $TN/(TN + FP)$, and Fall-out as $FP/(FP + TN)$.

| | Accuracy ↑ | Precision ↑ | Recall ↑ | Specificity ↑ | Fall-out ↓ |
|---|---|---|---|---|---|
| Easi3R (Chen et al., 2025) | 0.900 | 0.448 | 0.587 | 0.928 | 0.072 |
| WildGaussians (Kulhanek et al., 2024) | 0.353 | 0.437 | 0.893 | 0.931 | 0.069 |
| **Ours** | 0.885 | 0.255 | 0.244 | 0.940 | 0.060 |

Table 9: LPIPS metrics on the *T-3DGS* dataset.

Surprisingly, both Recall and Precision do not correlate with the quality of reconstruction or the quality of the TMR prompt. On the other hand, less common metrics like Specificity and Fall-out appear to be significantly more important.

Although overall accuracy suffers, T-3DGS proves to be a better prompt for SAM2 (Ravi et al., 2024) propagation, due to the lower false positive rate, which is both Specificity and Fall-out reflect.

## I MORE QUALITATIVE COMPARISONS

For qualitative comparison, we evaluate our method against SpotLessSplats (Sabour et al., 2024), Robust3DGS (Ungermann et al., 2024), and WildGaussians (Kulhanek et al., 2024). We provide corresponding renderings for the masks shown in the main paper in Sec. 4.2. Fig. 6 and 7 show reconstructions of several scenes from the *On-the-go* dataset on training and testing frames, respectively. Currently, most methods can produce fairly good reconstructions and avoid significant

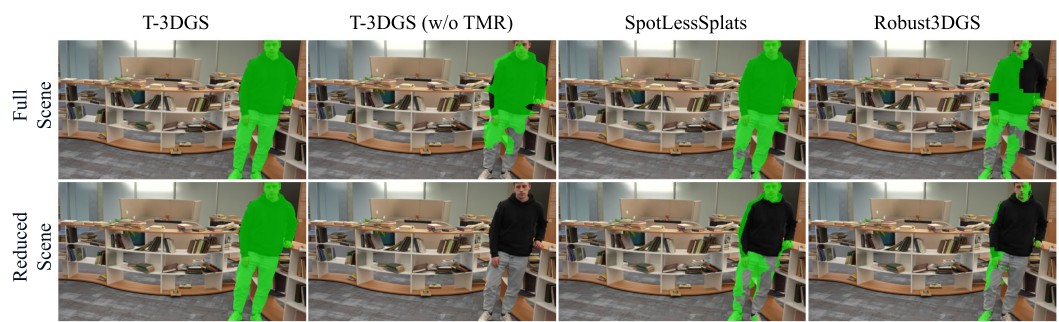

Figure 8: Comparison of predicted masks for full and reduced scenes.

artifacts, so generally, most methods produce fairly similar results (at least in the absence of semi-transient objects and other adversarial cases). Notably, compared to other residual-based methods, we avoid misclassifying high-frequency details and similar objects.

## J    HANDLING SEMI-TRANSIENT OBJECTS

Semi-transient objects have not been properly addressed in 3D scene reconstruction. Our method represents a significant improvement over previous work and can handle relatively complex scenarios. We provide details on the data capture process and the methodology employed for handling semi-transient objects. We also discuss the importance of both divergence estimation and mask propagation in handling semi-transient objects. Additionally, we discuss the limitations of our proposed method.

Our dataset includes two versions of some scenes: reduced and full. In reduced scenes, the camera operator moves from one end of the scene to the other. In full scenes, however, the operator retraces this path back to the starting position while semi-transient objects continue to move. As illustrated in Fig. 8, our proposed TMR module is essential for achieving good results in reduced scenes, which are particularly challenging. In full scenes, the additional frames lead to significantly improved mask predictions for all models because transient objects remain visible for longer periods. When fewer frames capture the scene, many methods mistakenly classify these transient objects as static. Overall, our findings highlight that effectively handling semi-transient objects is a major challenge in in-the-wild video processing. To develop the most challenging datasets and to rigorously compare different methods, it is important to consider not only the types of motion dynamic objects exhibit but also their movement relative to the camera.

As mentioned above, some of the scenes include semi-transient objects occluding the static scene for prolonged periods of time while remaining mostly still. As this period of time increases, semi-transient objects can effectively become static. Although this effect might seem irrelevant to the detection of dynamic objects, this is not the case. As shown in the Fig. 9, most methods mask the static background as if it were masking the semi-transient object. Notably, because WildGaussians relies heavily on semantic information, it can "propagate" the masks. However, this happens too late into the training process while our method avoids this problem, and this highlights the importance of using both divergence estimation and mask propagation algorithm we have proposed. Moreover, we aim to minimize false classifications of static objects as dynamic. As discussed earlier, even WildGaussians produces an excessive number of misclassifications for *TMR*. Therefore, our method is crucial for mask propagation to avoid introducing additional errors. This is in stark contrast to the competing methods, which have a lot more false positives. Mask propagation could introduce additional errors and might not contribute to overall quality improvement.

Our method reliably removes transient and semi-transient distractors and successfully reconstructs static artifact-free 3D scenes. However, we have observed that predicted masks tend to be inflated due to the low resolution of the extracted feature maps. Our method can also produce inconsistent results for small objects, as DINOv2 features are computed on patches. These problems could be addressed by using feature extractors with higher-resolution feature maps or guided upsampling.

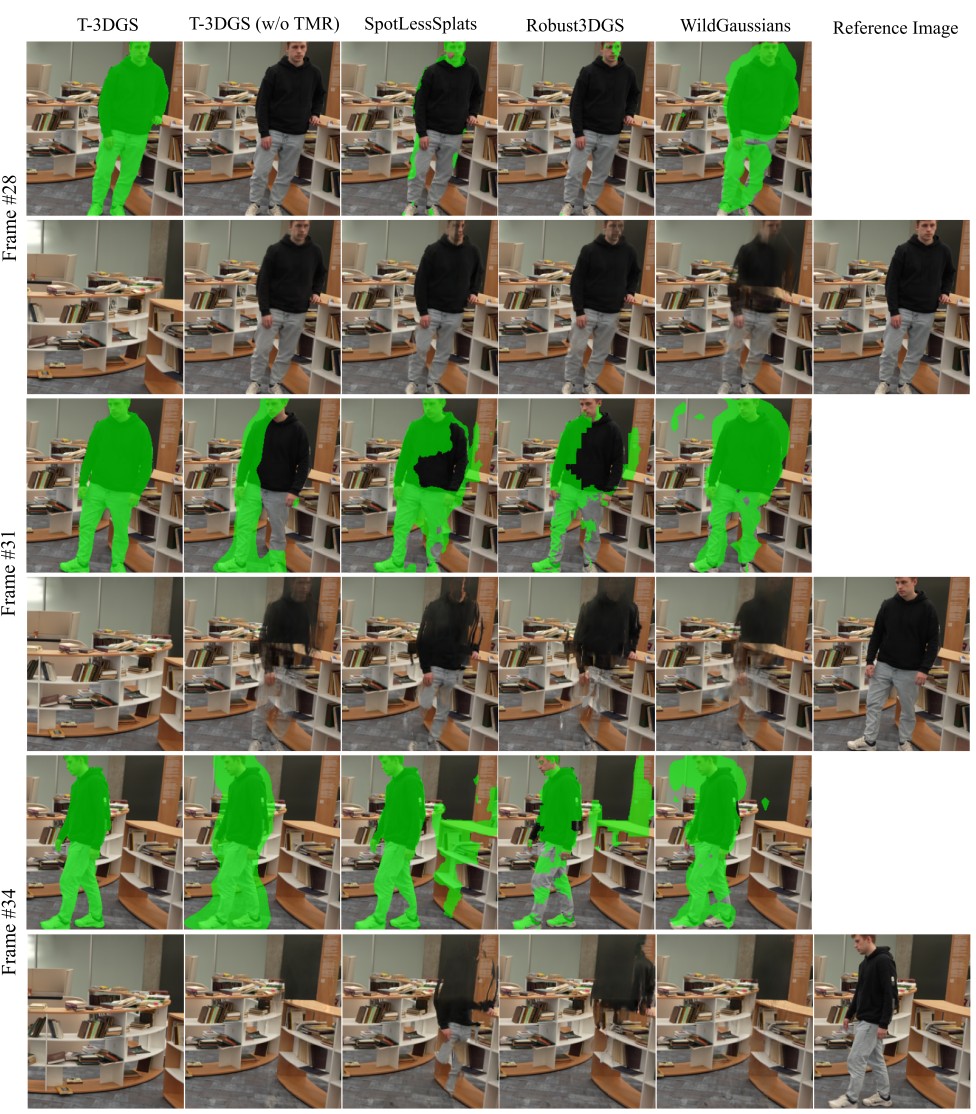

Figure 9: Comparison of predicted masks and scene reconstructions during the movement of semi-transient objects across different frames.

Additionally, the temporal refinement process is limited by a memory window of $N_m$ frames, which means that if an object disappears for more than $N_m$ frames and then reappears, it will be treated as a new instance with a different label. This can lead to inconsistent tracking and potentially affect the filtering process, especially for semi-transient objects that may temporarily leave the scene. Furthermore, our current filtering approach using global coverage scores may incorrectly filter out valid dynamic objects that undergo significant size changes, such as objects moving towards or away from the camera, or those experiencing perspective changes. We leave this aspect for future work.

