# OpenReview forum: "T-3DGS: Removing Transient Objects for 3D Scene Reconstruction"
_ICLR.cc/2026/Conference — Submitted to ICLR 2026_

### Official Review · Reviewer_nxFn · 2025-10-25

**Soundness:** 2
**Presentation:** 2
**Contribution:** 2
**Rating:** 4
**Confidence:** 4

**Summary:**

This paper introduces T-3DGS, which reconstructs static 3D scenes from monocular videos that contain transient objects. The pipeline begins with a Reconstruction Uncertainty Predictor (RUP) that uses DINOv2 features to produce a binary transient mask, which is then refined spatially with SAM and temporally with SAM2. During 3D Gaussian splatting, masked regions are excluded so optimization focuses on static content. The paper also introduces the T-3DGS dataset, a more challenging benchmark than prior sets. Experiments on three datasets show that T-3DGS achieves state-of-the-art performance.

**Strengths:**

1. This paper is easy to follow, the idea of detecting transient objects with feature extraction and refining them with SAM and SAM2 is reasonable and demonstrates good performance.
2. The proposed new dataset, T-3DGS, is more challenging than previous benchmark for evaluating models' performance on this task, and it is useful for the community.
3. Each contributions are properly evaluated through ablation studies.

**Weaknesses:**

1. My main concern is that this method is a combination of existing foundation models DINO, SAM and SAM2. It provides little novelty or inspirations, given these models are already well-explored in this community.
2. The comparison with Easi3R seems unfair, because as a feed-forward based method,  Easi3R is good for its inference speed and generality, and as optimization based method, the proposed method is expected to have better per-scene performance.
3. The TMR module comprises two components—SAM for spatial refinement and SAM2 for temporal refinement. What are the respective contributions of each to the refinement process, qualitatively or quantitatively?
4. Table 2 suggests most of the gains come from TMR. Since TMR leverages off-the-shelf SAM/SAM2, the novelty and impact of RUP are not convincingly demonstrated.

**Questions:**

1. Is the model structure of RUP introduced in supplementary material B?

---

> ### Author Response · Authors · 2025-11-21
>
> Thank you for your careful review and insightful comments.
>
> 1. Q: Is the model structure of RUP introduced in supplementary material B?
>
> Yes, the detailed architecture of the RUP module is provided in Supplementary Material Section B. For full transparency and reproducibility, we have also released the complete source code alongside the submission.
>
> 2. On the respective roles of SAM and SAM2 in TMR
>
> We have empirically discovered that SAM2’s temporal predictions are generally noisier than those of the original SAM, particularly over long sequences. In TMR, SAM2 is therefore used solely for efficient mask propagation, while the final per-frame refinement is performed with SAM. Applying SAM only once per sequence adds negligible overhead.
>
> 3. On comparison with Easi3R
>
> We included Easi3R because its paper explicitly positions itself as a fast, feed-forward alternative to optimization methods like ours. While the two paradigms have different strengths (speed vs. per-scene accuracy), Easi3R could in principle outperform optimization-based approaches on certain metrics or datasets. Including this recent and highly relevant baseline therefore provides a more comprehensive evaluation and anticipates potential reviewer questions.
>
> 4. On novelty and the impact of RUP (addressing concerns 1 and 4)
>
> While our method builds on strong foundation models (DINOv2, SAM, SAM2), its core novelty lies in the RUP module. This is evidenced by:
> (i) Tab. 5 showing that applying our TMR post-processing to other baselines sometimes can even degrade results;
> (ii) RUP alone already outperforming all prior work on sequences without semi-transient objects (e.g., On-the-go, Tab. 2);
> (iii) RUP’s ability to dramatically suppress false-positive dynamic labels (Fig. 4 and 5). TMR is merely an optional refinement for semi-transient cases - the primary advance and consistent SOTA performance are driven by RUP.
> (iv) While our approach can seem ad hoc and contrived, it’s a lot more justified and theoretically grounded. As discussed in L263-266, KL divergence is a robust technique that has proven itself in other applications.
> (v) Most methods require extensive hyperparameter tuning. However our method does not. We additionally performed sensitivity ablation of the KL threshold to prove it. (Table below reports average metrics on the On-the-go dataset for varying KL thresholds; 50 is used in the paper)
>
> |        | PSNR  | SSIM | LPIPS |
> |--------|-------|------|-------|
> | KL 15  | 23.38 | 0.80 |  0.18 |
> | KL 30  | 23.36 | 0.81 |  0.17 |
> | KL 50  | 23.41 | 0.81 |  0.17 |
> | KL 90  | 22.89 | 0.80 |  0.17 |
>
> We believe these clarifications and additional evidence clearly demonstrate the novelty and impact of our contributions. Thank you again for your constructive comments—we have incorporated several of your suggestions into the revised manuscript and supplementary material.

---

> > ### Comment · Reviewer_nxFn · 2025-11-27
> >
> > Thanks authors for their rebuttal. I agree that this work demonstrates better performance than existing works. However, the RUP module is heavily based on existing baselines, and the improvements come from using a stronger feature extractor and predicting more parameters to combine information from the residuals, without introducing novelty or instructive ideas into the community. Hence, I don't think it meets the high standard for ICLR, and I will maintain my current score.
> >
> > Typo: 3.1 section title Transient 2D Uncertainty Modeling is missing a period.

---

> > > ### Author Response · Authors · 2025-11-28
> > >
> > > Thank you for your reply. We fully agree that maintaining the high standards of ICRL is very important. However, we believe the statement that our method lacks novelty or instructive ideas is somewhat unfair. Although we build upon existing baselines, we do not merely rely on residuals in order to predict masks as done in prior work; instead, we substantially reinterpret these approaches in a novel way. In particular, Sec. 3.1 introduces novel algorithmic and conceptual components (in particular criterion (10)) which strongly deviate from currently explored methods. As evidenced by our qualitative and quantitative results, this enables us to address previously unexplored questions of priors in a principled way.
> > >
> > > Like WildGaussians, we utilize uncertainty modeling as our primary technique. However, our approach essentially has two distinct distributions - “background 3DGS” model and “GT frame” model. This separation enables a more robust probabilistic approach that explicitly treats the background model as a, in some sense, prior. In that sense, we detect outliers by calculating Bayesian surprise between posterior and prior distributions. In contrast, we could say that methods like WildGaussians essentially assume as little as possible about priors. The key observation is that the lack of strong priors results in numerous problems (e.g., rampant false positives and high sensitivity to hyperparameters). Introducing a separate background model as a prior offers a probabilistically meaningful solution.
> > >
> > > In other words, whereas (including WildGaussians) primarily detect regions that are difficult to reconstruct, our approach identifies regions that are surprisingly difficult given the background prior. KL divergence precisely quantifies this notion of "surprise" (i.e., how much the current frame deviates from the background prior), making it an ideal measure for our purpose.
> > >
> > > We believe this constitutes a previously unexplored approach that builds on a general and intuitively clear technique. In our view, this represents an important contribution that should not be understated.

---

### Official Review · Reviewer_Zdxa · 2025-10-27

**Soundness:** 2
**Presentation:** 2
**Contribution:** 2
**Rating:** 4
**Confidence:** 4

**Summary:**

This paper proposes T-3DGS, a framework for robust 3D scene reconstruction that removes transient and semi-transient objects from video sequences during 3DGS optimization. The key contribution is an unsupervised Reconstruction Uncertainty Predictor (RUP) that identifies transient distractors using multivariate uncertainty modeling with KL divergence and a Transient Mask Refiner (TMR) that enhances mask spatial and temporal consistency. Extensive experiments show that the proposed method outperforms baselines.

**Strengths:**

1. The paper is well motivated, addressing a practical problem in 3DGS-based scene editing.
2. The proposed T-3DGS dataset fills a gap for semi-transient object evaluation.

**Weaknesses:**

1. Limited novelty: The main contribution seems to be a combination of existing techniques (uncertainty estimation, semantic guidance, and SAM-based propagation), rather than addressing the deeper underlying issue of poor extrapolation and generalization in 3DGS.
2. Insufficient analysis: The paper lacks detailed sensitivity studies for key thresholds and hyperparameters, and the divergence-based uncertainty formulation remains largely heuristic without strong theoretical or empirical justification.
3. Dependence on SAM: Further evaluation is needed regarding the method’s reliance on SAM; segmentation noise or boundary inaccuracies may significantly influence transient detection and overall reconstruction performance.

**Questions:**

Does the SAM-based refinement introduce temporal artifacts or error propagation in long sequences?

---

> ### Author Response · Authors · 2025-11-21
>
> Thank you for your careful review and constructive feedback.
>
> 1. Q1: Does the SAM-based refinement introduce temporal artifacts or error propagation in long sequences?
>
> Our TMR refinement processes only a local temporal window of N nearest frames and does not perform global mask tracking. Therefore, long sequences can be processed incorrectly.
>
> An example of such a case would be a sequence where an object would remain static for a period of time, then disappear for at least N frames, then reappear and remain static.
> While cases like this are certainly a limitation, this would only be a problem when the object was not labeled as dynamic by RUP in the first place. But in this case, there are no clues present that would imply that the object is dynamic, which is beyond the scope of our paper.
>
> Additionally, global mask consistency across very long gaps is not required for accurate dynamic-object removal, so no temporal artifacts would be introduced.
>
> 2. On sensitivity analysis and justification of the divergence-based uncertainty
>
> We have added a new sensitivity study (Table below reports average metrics on the On-the-go dataset for varying KL thresholds; 50 is used in the paper):
>
>
> |        | PSNR  | SSIM | LPIPS |
> |--------|-------|------|-------|
> | KL 15  | 23.38 | 0.80 |  0.18 |
> | KL 30  | 23.36 | 0.81 |  0.17 |
> | KL 50  | 23.41 | 0.81 |  0.17 |
> | KL 90  | 22.89 | 0.80 |  0.17 |
>
> Performance is remarkably stable across a wide range, consistent with properties of KL divergence. Low sensitivity further validates our uncertainty formulation empirically.
>
> 3. On dependence on SAM
>
> As demonstrated by Tab. 5 in the supplementary material, we show that our method can produce the best initial prompts for SAM. While artefacts due to SAM itself exist, they are generally insignificant. Given that we also dilate our masks (which improves performance as per ablations), segmentation noise or boundary inaccuracies seem mostly irrelevant to the method presented in the paper.

---

> > ### Comment · Reviewer_Zdxa · 2025-11-24
> > **Reply to the rebuttal**
> >
> > Thanks for the rebuttal.
> > I sincerely acknowledge the author's efforts on the rebuttal. But I still believe the initial version of T-3DGS should be further refined, including more experiments and further illustrating the difference between T-3DGS and other baselines. In addition, the figures in the paper should be polished; for example, the font size in Figure 2 is too small to read.
> >
> > Thus, I tend to remain with the current score. Additionally, I encourage the author to try the proposed method on the feed-forward based 3DGS, since many works have investigated scene editing on per-scene 3DGS.

---

> > > ### Author Response · Authors · 2025-11-24
> > >
> > > We acknowledge the reviewer's opinion, but we would like to point out that these are new high-level critiques of our method that are not strongly related to the original review. Working on feed-forward-based 3DGS is an entirely new topic of research. We compared our method to all relevant baselines we are aware of, and can only compare to other methods based on specific suggestions.

---

### Official Review · Reviewer_dopq · 2025-10-31

**Soundness:** 3
**Presentation:** 3
**Contribution:** 2
**Rating:** 4
**Confidence:** 4

**Summary:**

This paper presents T-3DGS, a framework for reconstructing static 3D scenes from monocular video containing transient objects. T-3DGS is based on 3DGS, and introduces two key components: 1) reconstruction uncertainty predictor (RUP) that detects transient regions using semantic features and KL-divergence–based uncertainty modeling; 2) transient mask refiner (TMR) leveraging SAM/SAM2 for spatial refinement and temporal propagation. The combination allows 3DGS to focus on static content, which improves the reconstruction quality in real-world conditions. The authors conduct experiments on several datasets, and show better performance over existing methods.

**Strengths:**

1. The authors study an interesting problem of transient object, which is important for real-world scene reconstruction.
2. The method is well-designed and theoretically grounded.
3. The pipeline utilize DINOv2 features to provide robustness against color similarity and high-frequency textures.
4. The experiments compare against several baseline approaches and ablate key modules. The authors also introduce a new dataset with transient objects.
5. The authors provide qualitative results to clearly demonstrate better reconstructions.

**Weaknesses:**

1. The training pipeline is too heavy, which uses DINOv2 to extract features, use SAM for spatial refinement and SAM2 for temporal refinement.
2. Each submodule is adapted from existing techniques.
2.1 RUP uses DINOv2 features for semantic understanding, follows WildGaussians to build per-pixel residual with FeatUP and DSSIM, follows NeRF-W use uncertainty in 3DGS and separate static and transient objects.
2.2 The first part of the TMR uses SAM to clean up the noisy binary masks predicted by RUP.
2.3 The second part of TMR uses SAM2 to propagate masks.
3. The ablations do not fully separate the effects of KL, semantic feature, and TMR propagation.

**Questions:**

1. What happens if SAM or SAM2 produces incorrect boundaries?
2. What is the training interaction between RUP and 3DGS?

---

> ### Author Response · Authors · 2025-11-21
>
> Thank you for your valuable feedback and for your thoughtful comments.
>
> Q1: What happens if SAM or SAM2 produces incorrect boundaries?
>
> Incorrect boundaries from SAM / SAM2 do occur, but several aspects of our design mitigate their impact:
> - We explicitly optimize for a low false-positive rate in mask selection.
> - Using two-pass refinement with SAM2 generally improves boundary accuracy.
> - Additionally, because we aggregate masks across multiple frames, many ambiguities are resolved (e.g., selecting only a part of an object instead of the whole), and incorrect boundaries are propagated only when they appear consistently across frames, which results in very few failure cases overall.
>
> Moreover, precise mask boundaries are of little importance in our method. Tab. 3 shows that expanding the masks improves final performance, which indicates that small boundary inaccuracies have a negligible impact.
>
> Q2: Training interaction between RUP and 3DGS?
>
> As described in Sec. 3.1, in each training iteration we update both the 3DGS representation and the RUP module. However, this is not a joint optimization: the two components do not directly interact. Specifically, we detach the predicted masks when updating the 3DGS parameters and detach the rendered images when updating RUP. This detachment is crucial to prevent degenerate local minima and ensure stable training.
>
> W3: Ablations do not fully separate the effects of KL, semantic features, and TMR propagation
>
> To clearly separate the effects of different modules, we should start by considering WildGaussians. In some sense, this is the base model we started with - we simply train a model that predicts per-patch uncertainty, and masks are obtained with a simple uncertainty threshold. It's not paired like our method
>
> To extend this method to 2D we can consider different options, other than KL. A particularly interesting feature of KL is that it is not symmetric. We do not have to use it, and a more direct generalization of WildGaussians approach might include the sum of squared differences (L2) of uncertainties, which generalizes WildGaussians in a more direct way and is obviously symmetric. Although it is also not a metric because it violates the positivity axiom. We provide results for it (labeled “2D non-KL”). This approach performs close to our final method, but clearly, it is a little bit behind. We attribute it to probabilistic interpretations of KL divergence.
>
> To ablate choice of residuals - instead of using expensive upscaled features, we choose a more simple pair of residuals, e.g. DSSIM and RGB L1 residual. This method performs reasonably well but tends to mask static objects which is unacceptable for TMR propagation. This method can be interpreted as a lightweight version of our approach. (We can include additional qualitative comparisons in the final version of the paper).
>
> |                      | PSNR  | SSIM | LPIPS |
> |----------------------|-------|------|-------|
> |    Wild Gaussians    | 22.69 | 0.80 |  0.15 |
> |       2D non-KL      | 23.12 | 0.81 |  0.16 |
> |      2D L1+DSSIM     | 23.31 | 0.81 |  0.16 |
> | 2D DINO+DSSIM (final)| 23.41 | 0.81 |  0.17 |
>
>
> We already compared results with and without TMR for T-3DGS dataset. Additionally, in the supplementary material Tab. 5, we also reported results for WildGaussians with TMR, showing that RUP is essential for its proper functioning.

---

> ### Comment · Reviewer_dopq · 2025-11-28
>
> Thanks again for the rebuttal. I appreciate the authors’ clarifications, and the additional explanations provided. However, I believe there is room for improvement.
>
> My main concern remains that the paper seems to combine several existing techniques rather than offering a clear theoretical advancement. To meet the bar for ICLR, it would be helpful for the paper to push the theoretical aspect further and present deeper insight that clearly distinguishes the approach from prior work.
>
> Also, noting that TMR is ablated on the T-3DGS dataset does not fully address the question. Even if TMR is intended for semi-transient objects and the T-3DGS dataset is designed for these objects, this does not explain the absence of ablations on other standard datasets. These abaltions are important for demonstrating that incorporating TMR does not negatively impact performance on scenes without semi-transient objects.
> For these reasons, I will keep my current score.

---

### Official Review · Reviewer_Sz53 · 2025-11-01

**Soundness:** 3
**Presentation:** 3
**Contribution:** 3
**Rating:** 8
**Confidence:** 4

**Summary:**

This paper tackles the task of removing transient objects from a video. The transient objects break the assumption of static scenes that most 3D reconstruction approaches rely on. To mitigate this issue, the authors propose an uncertainty modeling-based approach to detect whether a mask is transient or not. Further, to ensure temporal consistency, the authors propose to use temporal refinement to enhance the mask quality. Experiments on various datasets demonstrate the effectiveness of the proposed approach.

**Strengths:**

- originality-wise: the idea of utilizing uncertainty modeling and mask propagation to handle dynamic objects is interesting.
- quality-wise: qualitative and quantitative results demonstrate the effectiveness of the proposed approach.
- clarity-wise: the paper is well-written in general.
- significance-wise: the problem of removing transient objects is important for downstream tasks of 3D reconstruction from in-the-wild videos.

**Weaknesses:**

1. For temporal refinement (L315): can we just use forward or backward propagation? How bad will the performance be qualitatively and quantitatively?

2. Can authors provide some runtime analysis?

3. How to determine the extent of dilation (L290) as it seems important from Tab. 3?

4. From the Fig. 2, it seems like RUP is not updated, which contradicts L170. Can authors clarify?

5. For Fig. 9, the T-3DGS's results do not seem to be from the same camera as the other methods or GT. Is this a bug or using the wrong one?

5. Please add a colorbar to Fig. 3. I am not sure which color means high uncertainty.

**Questions:**

See "Weakness"

---

> ### Author Response · Authors · 2025-11-21
>
> Thank you for the positive feedback and for the thoughtful comments and suggestions.
>
> 1. On the use of forward and backward passes
>
> Bidirectional propagation is a standard technique that improves temporal consistency. In our case, it additionally helps mitigate some SAM artifacts (e.g., over- or under-segmentation at object boundaries) and resolves ambiguities (such as whether the torso and legs belong to one or two objects). The final version of our paper will include additional qualitative examples in the supplementary material to better illustrate these benefits.
>
> 2. On the choice of dilation length
>
> As we use dilation in order to mitigate mask contraction during the training process (due to boundary patches having more accurately reconstructed static parts), the choice of dilation hyperparameter is due to two factors:
> - DINO operates on 14x14 patches that may cover both static and dynamic regions - i.e, there’s no reason to use dilation higher than 14.
> - Excessive dilation degrades reconstruction quality, especially for small objects (relative to the patch resolution).
>
> Since our primary concern is partially occluded dynamic objects, we empirically found a dilation of 10 pixels (a bit smaller than the DINO patch size of 14) to offer the best trade-off. The exact value is not critical; performance is stable in a range of values.
>
> 3. Clarifications and corrections
>
> We appreciate you pointing out the confusion regarding trainable modules. Indeed, only one RUM module is updated during training (indicated by the flame emoji in Fig. 2). Line 170 describes its training procedure. In the revised version, we will use a clearer notation (e.g., a different color or explicit “trainable” label) to avoid ambiguity.
>
> Figure 9 contains an error and will be corrected in the final version.
>
> For Figure 3, we omitted a colorbar because the absolute scale of the learned uncertainty values is somewhat arbitrary and less interpretable than the relative pattern. A textual description seemed more appropriate, but we are open to adding a colorbar if the reviewer believes it would improve clarity.
>
> Thank you again for your careful reading and valuable feedback.

---

### Author Response · Authors · 2025-12-03
**Final summary**

We would like to thank the reviewers for their valuable feedback. Below, we summarize the key updates in our rebuttal, addressing primary concerns and questions regarding empirical validation.

1) **Questions regarding novelty of the method.**
Several reviewers were concerned with the novelty of our method. We have clarified both algorithmic novelty and theoretical motivation. In particular we discussed how our approach relates to the concept of Bayesian surprise and how establishing an additional background model can resolve numerous issues that held back previous methods - in particular false positive detections and hyperparameter tuning.
We conclude that the primary issue lies not so much in novelty, but rather clarity of our presentation (e.g., in their question 4 reviewer Sz53 asked to clarify one of the essential aspects of our method). Moreover, reviewers have barely commented on RUP itself, which we found odd given its central role in the paper.

2) **Additional ablations on RUP.**
We performed additional ablations demonstrating exactly why our method performs better.  Using 2D formulation of uncertainty modeling we can address questions regarding choice of residuals - our method allows us to take advantage of both classical metrics like SSIM and novel ones that rely on features from ViT models like DINO. Moreover, we have demonstrated that using DINO cosine distance-based residuals (in place of something simple like RGB residuals) is also crucial.

3) **Concerns with respect to hyperparameter sensitivity.**
We performed an additional sensitivity ablation of the KL threshold that demonstrates robustness of our method. Unlike our competitors, that strongly rely on hyperparameter tuning (different hyperparameters for outdoors and indoors scenes, etc.), our method works sufficiently well for all scenes under the same hyperparameters. Moreover, this advantage is explained by theoretical aspects of our formulation. Instead of looking for absolute values of error, we instead rely on Bayesian surprise (as described above) further emphasizing both theoretical and empirical value of our work.

We hope that we addressed all of the reviewers' concerns. We believe these additions establish T-3DGS as a robust, practical advance for static 3D reconstruction in the presence of transient objects.

---

### Meta-Review · Area_Chair_xHvJ · 2025-12-08

**Summary:**

In the initial reviews, most reviewers raised concerns about the limited technical novelty and contribution of the work, insufficient experiments and comparisons, and missing studies. Three of the four reviewers, all of whom gave negative scores, participated in the discussion but chose to maintain their original ratings. They felt that the paper combines existing techniques without offering a sufficiently strong new contribution, and that the current presentation would require substantial revisions.

The AC has carefully read the paper, rebuttal, and discussions. While some issues were clarified, the core concerns about novelty, empirical validation, and presentation remain. Overall, the manuscript in its current form is not ready for publication at ICLR 2026.

**Reviewer Concerns:**

As mentioned above.

**Reviewer Scores:**

Reviewer Sz53: 8 (no response during rebuttal)
Reviewer dopq: 4 (replied but maintained)
Reviewer Zdxa: 4 (replied but maintained)
Reviewer nxFn: 4 (replied but maintained)

---

### Decision · Program_Chairs · 2026-01-26

Reject